# Bronchoscopic Endobronchial Valve Therapy for Persistent Air Leaks in COVID-19 Patients Requiring Veno-Venous Extracorporeal Membrane Oxygenation

**DOI:** 10.3390/jcm12041348

**Published:** 2023-02-08

**Authors:** Barbara Ficial, Stephen Whebell, Daniel Taylor, Rita Fernández-Garda, Lawrence Okiror, Christopher I. S. Meadows

**Affiliations:** 1Department of Adult Critical Care, Guy’s and St Thomas’ NHS Foundation Trust, St Thomas’ Hospital, Westminster Bridge Road, London SE1 7EH, UK; 2Intensive Care Unit, Townsville University Hospital, 100 Angus Smith Drive, Douglas, QLD 4814, Australia; 3Department of Thoracic Surgery, Guy’s and St Thomas’ NHS Foundation Trust, Guy’s Hospital, Great Maze Pond, London SE1 9RT, UK

**Keywords:** COVID-19, acute respiratory distress syndrome (ARDS), extracorporeal membrane oxygenation (ECMO), persistent air leak (PAL), bronchopleural fistula (BPF), endobronchial valve (EBV)

## Abstract

COVID-19 acute respiratory distress syndrome (ARDS) can be associated with extensive lung damage, pneumothorax, pneumomediastinum and, in severe cases, persistent air leaks (PALs) via bronchopleural fistulae (BPF). PALs can impede weaning from invasive ventilation or extracorporeal membrane oxygenation (ECMO). We present a series of patients requiring veno-venous ECMO for COVID-19 ARDS who underwent endobronchial valve (EBV) management of PAL. This is a single-centre retrospective observational study. Data were collated from electronic health records. Patients treated with EBV met the following criteria: ECMO for COVID-19 ARDS; the presence of BPF causing PAL; air leak refractory to conventional management preventing ECMO and ventilator weaning. Between March 2020 and March 2022, 10 out of 152 patients requiring ECMO for COVID-19 developed refractory PALs, which were successfully treated with bronchoscopic EBV placement. The mean age was 38.3 years, 60% were male, and half had no prior co-morbidities. The average duration of air leaks prior to EBV deployment was 18 days. EBV placement resulted in the immediate cessation of air leaks in all patients with no peri-procedural complications. Weaning of ECMO, successful ventilator recruitment and removal of pleural drains were subsequently possible. A total of 80% of patients survived to hospital discharge and follow-up. Two patients died from multi-organ failure unrelated to EBV use. This case series presents the feasibility of EBV placement in severe parenchymal lung disease with PAL in patients requiring ECMO for COVID-19 ARDS and its potential to expedite weaning from both ECMO and mechanical ventilation, recovery from respiratory failure and ICU/hospital discharge.

## 1. Introduction

Infection with SARS-CoV-2 virus leading to COVID-19 disease and respiratory failure is often associated with multiple air leak syndromes including pneumothorax, pneumomediastinum, pneumopericardium and subcutaneous emphysema [1,2,3]. Extensive parenchymal damage and pulmonary emboli may predispose to the development of necrotising pneumonia, lung cavitation and bronchopleural fistula (BPF) formation, with all leading to large, persistent air leaks (PAL). The risk is increased in those receiving invasive mechanical ventilation (IMV), with recent data suggesting an incidence as high as 15–40% [4,5,6,7] versus 6–10% in non-COVID acute respiratory distress syndrome (ARDS) patients [8,9,10]. A similar pathology was also observed in cases of SARS-CoV-1 and Middle East respiratory syndrome (MERS) [11,12,13], suggesting it is a common feature of severe coronavirus respiratory illness. The pathophysiological mechanisms behind this are not fully understood; however, the subpleural predilection of pulmonary disease, airway distortion and bronchial wall friability may be implicated [14,15]. Additionally, high respiratory drive and associated increased transpulmonary mechanical stress may explain the presence of pneumothoraces in COVID-19 patients on first medical contact [16,17]. Multiple chest drains are often required to drain pneumothoraces and allow lung expansion. The combination of stiff, poorly compliant lungs from ARDS and large air leaks can make ventilation difficult with insufficient minute ventilation (MV) to sustain adequate gas exchange.

Veno-venous extracorporeal membrane oxygenation (VV-ECMO) for COVID-19 ARDS provides a non-pulmonary route for gas exchange in those with reduced MV and intractable air leaks and permits lung rest by allowing a reduction in mechanical power, thus, minimising ventilator-induced lung injury (VILI). According to international registry data, in-hospital mortality for ECMO-supported COVID-19 patients ranges from 36.9 to 51.9% with experienced, higher-volume centres demonstrating more favourable outcomes [18].

Poor lung healing results from a combination of steroid treatment, super-added bacterial pneumonia with cavitation and persistent air leaks maintaining fistulae, resulting in a failure to wean both ECMO and ventilation. Traditional means of managing air leaks with pleurodesis and surgical resection of damaged lungs are often not feasible because of incomplete lung expansion (making apposition of the lung to the chest wall difficult) and inability to tolerate single-lung ventilation, respectively.

Bronchoscopic endobronchial valve treatment has been successfully used for the treatment of air leaks from multiple aetiologies with good results [19,20,21,22,23]. Data on the use of EBV for ventilated patients are limited, and their use in patients on ECMO has only been reported sporadically [24,25,26]. This series reports our experience of EBV in the management of intractable air leaks in patients on ECMO for severe respiratory failure secondary to COVID-19 disease.

## 2. Materials and Methods

This is a retrospective review of a consecutive series of patients with large air leaks related to SARS-CoV-2 infection. All patients were admitted to the high-volume specialist ECMO intensive care unit (ICU) at Guy’s and St Thomas’ Hospitals in London with severe respiratory failure requiring VV-ECMO. Standard lung rest settings at admission were mandatory pressure control ventilation, with driving pressure of 10 cm H_2_O, PEEP ≤ 10 cm H_2_O and a respiratory rate of 10. Patients were considered for endobronchial valve (EBV) placement if there were persistent (>10 days), large air leaks (>500 mL/min) despite conventional therapy (optimal pleural drainage, reduced mechanical ventilator pressure, lung rest, attempted recruitment of collapsed lung segments, and isolated lung ventilation and pleurodesis where indicated). Multidisciplinary discussion between teams from ICU, respiratory medicine and thoracic surgery had deemed that no other methods of management of the air leaks were feasible, including lung surgery.

EBV deployment was performed in all cases at the bedside in ICU (see Figure 1). A 2.8 size fibreoptic bronchoscope was used via the endotracheal or tracheostomy tube. Selection for the site of EBV placement was assessed using sequential balloon occlusion of lobar then segmental bronchi while observing for immediate cessation of air leak at the pleural drains. Once the source of the air leak was identified, an appropriate-sized Zephyr^®^ endobronchial valve (4.0–5.5 mm, Pulmonx, Redwood City, CA, USA) was deployed to occlude the segment or lobe. Consent for the procedure was obtained as per hospital policy in adults who lacked capacity to consent to investigations or treatment, and risks and benefits were discussed with patients’ families. All procedures were performed by a thoracic surgeon experienced in the procedure. Maintained cessation of air leaks was subsequently confirmed with the absence of both bubbling from an under-water seal pleural drain system and measured minute volume leak on the ventilator.

Patient characteristics and clinical data were collected retrospectively from a prospectively maintained patient electronic health record (Intellispace Critical Care and Anaesthesia, Philips Healthcare, Koninklijke, Amsterdam, the Netherlands). Imaging data were collected from an electronic radiology reporting system (PACS, Sectra Medical, Linköping, Sweden). Continuous data were reported as mean and standard deviation or range, and categorical data were reported as count and percentage. Survival was measured from the date of hospital discharge. Ethical approval was sought and waived on the basis that this was a retrospective study.

## 3. Results

### 3.1. Patient Characteristics and Complications

Between March 2020 and March 2022, a total of 152 patients required VV-ECMO for COVID-19-related ARDS. Of these patients, 10 had EBV inserted for persistent, large air leaks. Demographics and clinical characteristics at admission are shown in Table 1. The mean age was 38.3 years (range 27–52), and six patients (60%) were male. Five patients (50%) had zero pre-existing co-morbidities. All patients were treated with evidence-based therapies for COVID-19, including corticosteroids [27]. All experienced concurrent secondary clinical issues, including pneumothorax and pulmonary infection, and five (50%) had pleural space infection ipsilateral to the air leak. Gram-negative and fungal organisms predominated, and, where present, isolates from the pleural space were identical to those identified in pulmonary samples. Of note, six (60%) patients had cytomegalovirus (CMV) reactivation with positive IgG and detectable viral load in blood. Nine (90%) had thrombotic complications, predominantly pulmonary emboli, and all underwent surgical procedures during the ICU admission, including intercostal pleural drainage and percutaneous tracheostomy formation. Three (30%) patients in this series suffered significant intrathoracic bleeding, requiring emergent thoracostomy intervention and one endovascular embolisation procedure to achieve haemostasis. Two patients (20%) underwent emergency caesarean section during their COVID-19 illness.

### 3.2. Characteristics of Air Leaks and Endobronchial Valve Deployment

The characteristics of the air leaks, procedural details, clinical course and outcomes are summarised in Table 2. The air leaks occurred at an average of 14.5 (+/−15.6) days following the initiation of IMV. In two (20%) cases, the leak occurred prior to intubation whilst the patients were receiving non-invasive respiratory support. In nine (90%) cases, the persistent air leak was unilateral, and one case had bilateral leaks, occurring consecutively. In total, eight (73%) air leaks arose from the right lung, of which four (50%) originated from the lower lobe and three (38%) from the middle lobe. The left lung air leaks (27% of total) occurred in the lower lobe (66%) and from the lingula and upper lobe (33%). An average of 2.3 (range, 1–3) EBVs were deployed per patient. The valve placement procedure occurred at a mean of 18.5 days following the onset of the air leaks, and in all cases, the cessation of the air leak was achieved with no immediate complications. Nine (90%) patients were receiving VV- ECMO support at the time of the valve deployment procedure; one patient was receiving IMV, having developed an air leak 3 weeks post ECMO decannulation. The mean duration of ECMO support for respiratory failure was 42.6 days (range, 15–82 days), significantly longer than that reported for COVID-19-related severe respiratory failure [28]. In four (40%) patients, the air leak subsequently recurred at a lower volume but, in all cases, was not sufficient to prevent the weaning of ECMO and ventilatory support. All pleural drains were removed in survivors following continued cessation of air leak or elimination of significant leak at a mean of 24.1 days (range, 4–70 days) following placement of valves; time to drain removal was shorter in those patients without recurrence of air leak (average, 12.5 days; range, 4–28 days).

### 3.3. Imaging and Patient Outcomes

Imaging before and after EBV placement for selected cases is shown in Figure 2 (imaging for all cases in Appendix A). Cross-sectional (computed tomography (CT)) images were acquired for all patients prior to EBV placement. A single CT slice was selected here to optimally demonstrate the site of persistent air leak and co-existing pulmonary pathology. The images also visually illustrate the reasons for the failure of conventional IMV and the weaning of extracorporeal support. Further imaging, both plain film and cross-sectional, in the post-valve deployment phase show the resolution of the air leaks and the degree of normalisation of lung parenchyma prior to discharge from hospital.

Eight (80%) patients survived to hospital discharge and were alive at the most recent follow-up period (range, 3–12 months following hospital discharge). Two patients died from progressive pulmonary parenchymal disease and multi-organ failure. At follow-up, six survivors had persistent, reduced exercise tolerance compared to the pre-morbid baseline, with three requiring ambulatory oxygen therapy. Two patients had regained good exercise tolerance post-discharge.

## 4. Discussion

To our knowledge, this is the first case series of EBV placement for PAL in patients requiring VV-ECMO for COVID-19. The patients presented in this series all had severe respiratory failure, as defined by the commissioning criteria for ECMO [29]. Despite optimal evidence-based management of COVID-19 disease, all cases had failed conventional ventilatory management and required extracorporeal support. The duration of ECMO in this series (42.6 days) significantly exceeds the ECMO duration for COVID-19 by a factor of three in a systematic review (15.8 days) [28]. Therefore, all patients exhibited the consequences of prolonged COVID-19 critical illness, ventilation and ECMO, including musculoskeletal deconditioning, delirium, recurrent episodes of secondary pulmonary infection and immunocompromise from steroid use.

Air leak syndromes have emerged as a common feature of severe COVID-19 ARDS. Preferential peripheral and subpleural location of the disease, lung stress through high spontaneous MV (patient self-inflicted lung injury), hyperinflammation and in situ thrombosis are thought to be contributing factors [11,30,31]. Whilst the majority of air leaks resolve with conservative management, the patients presented here had refractory PAL precluding weaning from ECMO or mechanical ventilation. 

Traditional open resective surgical approaches to PAL in this cohort of ECMO patients are unattractive for several reasons. Firstly, bleeding complications may arise from several mechanisms despite meticulous monitoring of systemic anticoagulation. Bleeding is well described in patients receiving VV-ECMO therapy [32] primarily through circuit-induced coagulopathy, with this risk further increased in cases requiring fibrinolytic therapy for PE. Secondly, surgical repair necessitating lung resection in the setting of prolonged critical illness could limit respiratory reserve and, thus, the ability to wean from ECMO. Thirdly, surgery in the presence of pulmonary and/or pleural infection would likely lead to prolonged tissue inflammation, reduced defect healing and significantly reduced lung parenchymal integrity. Super-added bacterial infection has been reported relatively infrequently in a general COVID-19 population [33], but it is more common in mechanically ventilated patients, with adverse consequences. Finally, certain common patient characteristics, including steroid use, poor nutritional state, immunosuppression and deconditioning, favour a non-surgical approach in this patient group.

Endoscopic techniques are a minimally invasive alternative for the definitive treatment of air leaks in selected patients where conventional management has failed. Whilst previously employed for lung volume reduction in patients with emphysema, endobronchial one-way valves have since been used in cases of PAL in self-ventilating patients. Travaline and colleagues demonstrated that it is an effective and minimally invasive strategy for prolonged air leaks in 40 self-ventilating patients [34]; moreover, when complete leak cessation was not achieved, the significant reduction in leak volume was still beneficial. Evidence for EBV use remains sparse, however, especially in mechanically ventilated [35,36] and ECMO populations [24,25,26,37,38].

In our series, EBV placement via a bronchoscopic approach appeared to be safe and enabled immediate ventilatory lung recruitment and subsequent weaning from ECMO support. In those for whom the air leak did not recur, intercostal drain removal shortly followed valve deployment. If air leak recurred, leak volume was substantially lower than prior to EBV placement but did lead to a markedly longer period on ECMO and a longer interval until chest drain removal. Nevertheless, in this small series, 80% of patients survived to both ICU and hospital discharge. The degree of longstanding pulmonary pathology, dependence on ambulatory oxygen and reduced exercise tolerance in many is an indicator of the severity of their disease.

## 5. Conclusions

EBV placement for PAL in patients with severe parenchymal lung disease secondary to COVID-19 appears to be feasible and safe. In complex patients, where conventional management, including ECMO, has failed and surgical intervention presents an intolerably high risk, bedside EBV placement can be a successful strategy to enable patients to recover to hospital discharge.

## Figures and Tables

**Figure 1 jcm-12-01348-f001:**
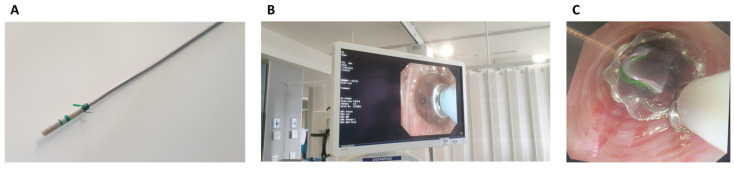
Endobronchial valve placement at the bedside in the intensive care unit. (**A**) Zephyr endobronchial valve prior to deployment; (**B**) selective endobronchial balloon occlusion in progress; and (**C**) endobronchial valve deployed in the airway.

**Figure 2 jcm-12-01348-f002:**
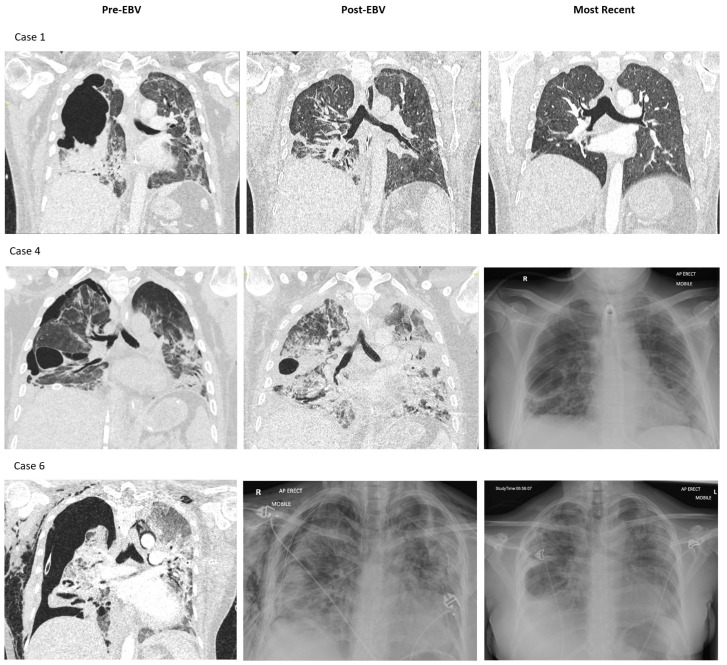
Imaging of selected cases before endobronchial valve (EBV) placement, after EBV placement and the most recent imaging available. CT imaging was used where available and plain film X-ray presented otherwise.

**Table 1 jcm-12-01348-t001:** Demographics and admission characteristics.

	All (*n* = 10)	Case 1	Case 2	Case 3	Case 4	Case 5	Case 6	Case 7	Case 8	Case 9	Case 10
Sex	60% Male	Male	Female	Female	Male	Male	Female	Male	Male	Female	Male
Co-morbidities	-	Nil	Diverticulosis; GORD; anxiety; obesity	Nil	Nil	Pemphigus	Nil	Nil	Obesity; depression	Pre-eclampsia	Obesity; type 2DM
Primary problem	-	COVID-19	COVID-19	COVID-19	COVID-19	COVID-19	COVID-19	COVID-19	COVID-19	COVID-19	COVID-19
Concurrent problems	-	Massive PE; right haemothorax	Tubo-ovarian abscess; right basal segmental PE; left hydropneumothorax; right pneumothorax	Right pneumothorax; haemothorax; CMV reactivation	Right tension pneumothorax; loculated hydro-pneumothorax; CMV reactivation	Subsegmental PE; right haemothorax; CMV reactivation	Lower limb DVT; right pneumothorax; CMV reactivation	HIT; right apical pneumothorax; CMV reactivation	Bilateral multiple PE; renal, splenic, liver embolic infarcts; right and left pneumothorax	Bilateral multiple PE; multiple DVT; radial artery thrombus; CMV reactivation	Right tension hydro-pneumothorax; multiple PE left and right lower lobe segments; RV thrombus
Interventions	-	Thoracotomy; pleural drainage; percutaneous tracheostomy	Percutaneous tracheostomy; pleural drainage	Thoracotomy; single-lung ventilation; pleural drainage	Pleural drainage (CT-guided); intrapleural thrombolysis	Pleural drainage; endovascular coiling distal right phrenic artery	Emergency caesarean section; pleural drainage	Pleural drainage; percutaneous tracheostomy	Pleural drainage; percutaneous tracheostomy	Emergency caesarean section; pleural drainage; percutaneous tracheostomy	Pleural drainage; percutaneous tracheostomy; single-lung ventilation; pleural irrigation
Secondary pulmonary infection	10 (100%)	*K aerogenes; Prevotella oralis*	*P aeruginosa; E faecium*	*P aeruginosa; S marcescens; C striatum*	*C albicans; C koseri*	*C albicans; Acinetobacter nosocomialis; P aeruginosa*	*C albicans; P mirabilis*	*E cloacae; Aspergillus fumigatus; E faecium; S marcescens*	*S.aureus; Fusobacterium nucleatum; Prevotella nanceiensis; C albicans*	*S.aureus; E.coli*	*S.aureus; C albicans; Kytococcus schroeteri*
Secondary pleural infection	5 (50%)	*K aerogenes*; *Prevotella oralis*	*P aeruginosa; E faecium; E faecalis*	*P aeruginosa; C striatum*	*No*	*C albicans; E faecium*	No	No	No	*E.coli*	No

PE—pulmonary embolism, HIT—heparin-induced thrombocytopaenia, CMV—cytomegalovirus, GORD—gastro-oesophageal reflux disease, DM—diabetes mellitus, RV—right ventricle, VV-ECMO—veno-venous extracorporeal membrane oxygenation, DVT—deep vein thrombosis, IMV—invasive mechanical ventilation, and CT—computed tomography.

**Table 2 jcm-12-01348-t002:** EBV procedural details, air leak characteristics, clinical course and outcomes.

	All (*n* = 10)	Case 1	Case 2	Case 3	Case 4	Case 5	Case 6	Case 7	Case 8	Case 9	Case 10
Site of EBV placement	-	Right lower lobe basal segments	Left lower lobe posterior and lateral basal segments	Right lower lobe apical and basal segments	Right middle lobe bronchus	Right lower lobe basal segments	Right middle lobe bronchus	Right upper lobe all segments	Right middle lobe; apical and posterior basal segments of left lower lobe	Lingula, both segments; left upper lobe	Right lower lobe anterior, medial and lateral basal segments
Numbers of EBV (size)	2.4 ± 0.8	2 (5.0 mm and 4.0 mm)	2 (4.0 mm and 5.5 mm)	3 (4.0 mm)	1 (5.5 mm)	3 (4.0 mm)	1 (5.5 mm)	3 (4.0 mm)	3 (5.5 mm and 2 × 4.0 mm)	3 (2 × 4.0 mm and 5.5 mm)	3 (4.0 mm)
Cessation of leak at end of procedure	10 (100%)	Yes	Yes	Yes	Yes	Yes	Yes	Yes	Yes	Yes	Yes
Immediate complications	0 (0%)	Nil	Nil	Nil	Nil	Nil	Nil	Nil	Nil	Nil	Nil
VV-ECMO at time of EBV procedure	9 (90%)	Yes	No	Yes	Yes	Yes	Yes	Yes	Yes	Yes	Yes
Days from initial IMV to air leak	14.5 ± 15.6	35	43	3	−1	11	24	−1	9	21	1
Days from air leak to EBV	18.5 ± 14.0	18	11	28	19	29	15	39	25	16	15
Days of ECMO	42.6 ± 19.6	43 *	15	82	43	61	39 *	43	52	22	26
Recurrence of air leak	4 (40%)	No	No	Yes	No	Yes	No	Yes	No	No	Yes
Days to removal of all pleural drains post EBV	24.1 ± 23.5 ^$^	4	7	70	28	-	10	37	-	13	†
Alive (follow up time)	8 (80%)	Yes (>1 year)	Yes (>1 year)	Yes (>1 year)	Yes (>1 year)	No	Yes (6 months)	Yes (6 months)	No	Yes (3 months)	Yes (3 months)
Functional status		Mildly reduced ET	Good ET	Reduced ET; ambulatory oxygen	Good ET	-	Reduced ET; ambulatory oxygen	Reduced ET; ambulatory oxygen	-	Reduced ET	Mildly reduced ET

EBV—endobronchial valve; ECMO—extracorporeal membrane oxygenation; VV-ECMO—veno-venous extracorporeal membrane oxygenation; ET—exercise tolerance; * combined duration of ECMO with two separate runs during the same admission; ^$^ calculated for survivors and non-missing data; and ^†^ information not available.

## Data Availability

The data presented in this study are available in Table 1 and Table 2 and Appendix A.

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
