# Peer review of "Bronchoscopic Endobronchial Valve Therapy for Persistent Air Leaks in COVID-19 Patients Requiring Veno-Venous Extracorporeal Membrane Oxygenation"

_jcm, 2023, doi:10.3390/jcm12041348_

Round 1
Reviewer 1 Report
This is a well written review of a useful technique. The images presented are excellent. The authors are to be congratulated for their excellent clinical care. I would like to invite the authors to include details of the positive pressure ventilation being received by their ECMO patients and whether they considered extubating their patients on ECMO to reduce their airleaks?
Author Response
Reviewer 1
- I would like to invite the authors to include details of the positive pressure ventilation being received by their ECMO patients and whether they considered extubating their patients on ECMO to reduce their airleaks?
Authors’ response
Our approach to ventilation in a newly cannulated ECMO patient is to allow a degree of lung rest: our standard ‘lung rest’ settings are mandatory pressure control mode with driving pressure 10 cmH2O, PEEP 10 cmH2O, respiratory rate 10; this normally results in minimal or absent native lung ventilation, and ultra-protective lung ventilation (TV 3-4ml/kg IBW) when compliance improves. If oxygenation via ECMO support only proves difficult, we would opt for higher ventilatory pressures within the limits of protective lung ventilation (TV 6ml/kg IBW, Plateau pressure <30 cmH2O, Driving pressur
e <14 cmH2O).
We did consider extubation and awake ECMO and we managed to achieve it eventually in case 1 and 6. In other patients it was discounted for multiple factors, including but not limited to: work of breathing, secretion burden, sepsis and high metabolic demands, delirium.
We thank the reviewer for this suggestion and have included a description of our ‘lung rest’ settings in the Materials and Methods section [ line 79-81 ]
Reviewer 2 Report
Dear authors,
The COVID-19 pandemic has led to an unprecendeted increase in the number of rare complications such as persistent air leak (PAL) and data on their management is highly wanted. I read your manuscript with interest.
Abstratc: Well written.
Introduction: Well written.
Materials and Methods: Well described.
Results / Discussion: -Any comments on the high pleural infection incidence in the study group compared to the literature? Was the pleural infection prior or after the PAL and could this play a role?
-9 patients were receiving vv-ECMO when EBVs were placed. The remaining one received ECMO after EBV placement i assume. Was that related to the PAL?
- Please correct second Table's title to Table 2 (line 170)
-Is there a plan to remove valves at some point? Is there literature on the long-term effects of EBVs for PAL (in non-COVID patients probably)?
Overall: Interesting topic and well presented.
Best regards.
Author Response
Reviewer 2
- Any comments on the high pleural infection incidence in the study group compared to the literature? Was the pleural infection prior or after the PAL and could this play a role?
Authors’ response
We agree with the reviewer’s assessment, certainly the high incidence of pleural infection stands out. However, this is a highly selected group of patients with bronchopleural fistulae.
In most cases we had not obtained pleural fluid samples prior to the air leak occurring, therefore it is difficult to establish a temporal link; but agree that infections could represent both cause and consequence of BPF. Other factors potentially contributing to high pleural infection rate: multiple pleural instrumentation and prolonged drain permanence, significant intrathoracic bleeding (3 patients), immunosuppression from steroid use. Finally, we cannot exclude that the increased workload during the pandemic had an impact on healthcare associated infections.
Reviewer 2
- 9 patients were receiving vv-ECMO when EBVs were placed. The remaining one received ECMO after EBV placement I assume. Was that related to the PAL?
Authors’ response
We thank the reviewer for pointing this out, as it was not specified in the manuscript.
One patient (case 2) developed an air leak 3 weeks post ECMO decannulation, as a result of evolving necrotising pneumonia; they required re-intubation and subsequent EBV placement. We have revised the manuscript to include this information [ line 146-147 ]
Reviewer 2
- Is there a plan to remove valves at some point? Is there literature on the long-term effects of EBVs for PAL (in non-COVID patients probably)?
Authors’ response
Valve removal was organised by our thoracic team once the patients had been discharged home and had been clinically stable for some time; on average, valves were removed between 6-9 months post ICU discharge. We did not include time to EBV removal as this was also influenced by non-clinical factors (timing of follow-up appointments, planning and scheduling of elective hospital admission for valve removal).
Literature on long-term effects is limited: in one of the biggest retrospective studies on EBVs for PAL (Fiorelli et al, reference 19), valves were removed in 82% of cases after 13483 days; no recurrence of air leak was reported following valve removal, and no issues were reported for patients whose valves were not removed (average follow-up was 37 months).